# Modified gemcitabine, S-1, and leucovorin combination for patients with newly diagnosed locally advanced or metastatic pancreatic adenocarcinoma: A multi-center retrospective study in Taiwan

Chia-Yu Chen[1], Shih-Hsin Liang[1,2], Yung-Yeh Su[3,4], Nai-Jung Chiang[3,4], Hui-Ching Wang[5], Chang-Fang Chiu[1,6,7], Li-Tzong Chen[3,4,5☯*], Li-Yuan Bai [iD][1,6☯*]

1 Division of Hematology and Oncology, Department of Internal Medicine, China Medical University Hospital, Taichung, Taiwan, 2 Department of Internal Medicine, Dalin Tzu Chi Hospital, Buddhist Tzu Chi Medical Foundation, Chia-Yi, Taiwan, 3 National Institute of Cancer Research, National Health Research Institutes, Tainan, Taiwan, 4 Division of Hematology and Oncology, Department of Internal Medicine, National Cheng Kung University Hospital, College of Medicine, National Cheng Kung University, Tainan, Taiwan, 5 Department of Internal Medicine, Kaohsiung Medical University Hospital, Kaohsiung Medical University, Kaohsiung, Taiwan, 6 College of Medicine, School of Medicine, China Medical University, Taichung, Taiwan, 7 Cancer Center, China Medical University Hospital, Taichung, Taiwan

☯ These authors contributed equally to this work.
* leochen@nhri.edu.tw (L-TC); lybai6@gmail.com (L-YB)

## Abstract

### Background

In pancreatic cancer, toxicities associated with current chemotherapeutic regimens remain concerning. A modified combination of gemcitabine, S-1, and leucovorin (GSL) was used as the first-line treatment for newly diagnosed locally advanced or metastatic pancreatic adenocarcinoma patients.

### Methods

GSL was administered every 2 weeks—intravenous gemcitabine 800 mg/m$^2$ at a fixed-dose rate of 10 mg/m$^2$/min on day 1 and oral S-1 (80–120 mg/day) plus leucovorin 30 mg twice daily on days 1–7. We retrospectively analyzed the feasibility of GSL and patient outcomes in three medical centers in Taiwan.

### Results

Overall, 49 patients received GSL with a median follow-up of 24.9 months from May 2015 to March 2019. The median patient age was 68 years (range, 47–83 years), with a marginally higher number of females (57.1%). Among the 44 patients who underwent image evaluation, 13 demonstrated a partial response (29.5%) and 17 presented with stable disease (38.6%). The partial response rate and stable disease rate was 26.5% and 34.7%, respectively, in the intent-to-treat analysis. The median time-to-treatment failure was 5.79 months

**Data Availability Statement:** All relevant data are within the paper and its Supporting Information files.

**Funding:** LYB: grants from the National Health Research Institutes, Taiwan (NHRI-109A1-CACO-13202002), the Ministry of Health and Welfare, Taiwan (MOHW109-TDU-B-212-134026), and Ministry of Science and Technology, Taiwan (MOST109-2321-8-006-011-) during the study. CFC: grants from the Ministry of Health and Welfare, Taiwan (MOHW109-TDU-B-212-010001).

**Competing interests:** The authors declare no conflict of interest.

(95% C.I., 2.63–8.94), progression-free survival was 6.94 months (95% C.I., 5.55–8.33), and overall survival time was 11.53 months (95% C.I., 9.94–13.13). For GSL treatment, the most common grade 3 or worse toxicities were anemia (18.3%), neutropenia (6.1%), nausea (4.1%), and mucositis (4.1%). Treatment discontinuation was mostly due to disease progression (65.3%).

## Conclusions

The modified GSL therapy can be a promising and affordable treatment for patients with advanced and metastatic pancreatic cancer in Taiwan. A prospective trial of modified GSL for elderly patients is currently ongoing in Taiwan.

## Introduction

Pancreatic cancer is an aggressive and lethal cancer owing to its late presentation and resistance to chemotherapy. Gemcitabine has been considered the reference treatment since Burris documented that gemcitabine resulted in a greater clinical benefit response (23.8% vs. 4.8%) and prolonged survival (5.65 vs. 4.41 months) [1]. Several gemcitabine-based combinations or non-gemcitabine-containing regimens failed to confirm the superiority till the trials of FOLFIRINOX and nab-paclitaxel plus gemcitabine [2,3]. Although there is a clear benefit in using these combinations over gemcitabine alone, the prognosis of patients with advanced or metastatic pancreatic cancer remains poor, with a median overall survival of 8 to 11 months and an estimated 2-year survival of only 2%.

Notably, the modest improvement in overall survival achieved with FOLFIRINOX is accompanied by considerable toxicities. For instance, although filgrastim was allowed for high-risk patients, the incidence of grade 3 or 4 neutropenia and fatigue was reportedly 45.7% and 23.6% in patients treated with FOLFIRINOX, respectively [2]. The toxicity profiles are even greater in Asian patients, leading to several modifications to FOLFIRINOX. For example, a phase II prospective trial evaluated modified FOLFIRINOX (intravenous oxaliplatin 85 mg/$m^2$, irinotecan 150 mg/$m^2$, 5-fluorouracil infusion 2400 mg/$m^2$ over 46 h) without prophylactic pegfilgrastim in patients with metastatic pancreatic cancer. Despite the modified schedule, the incidence of grade 3 or higher neutropenia was 47.8% [4]. In the MPACT trial using nab-paclitaxel plus gemcitabine, the incidence of grade 3 or 4 neutropenia, fatigue, and peripheral neuropathy was 38%, 17%, and 17%, respectively [3]. Although the combination of nab-paclitaxel and gemcitabine appeared less toxic, this regimen was not reimbursed by Taiwan's National Health Insurance before 2020. This situation implies an unmet medical need to provide a feasible and affordable therapeutic option for patients with pancreatic cancer in Taiwan.

S-1 is an oral fluoropyrimidine preparation containing tegafur, gimeracil, and oteracil. In Taiwan, this preparation is reimbursed for patients with locally advanced unresectable or metastatic pancreatic cancer. In the phase III GEST trial in patients with locally advanced or metastatic pancreatic cancer, S-1 (80 to 120 mg daily on days 1 through 28 every 42 days) was compared with gemcitabine monotherapy in a non-inferiority design, and S-1 plus gemcitabine (GS; gemcitabine 1000 mg/$m^2$ on days 1 and 8 plus S-1 30 mg to 50 mg orally twice daily on days 1 through 14 of a 21-day cycle) was compared with gemcitabine monotherapy in a superiority design [5]. S-1 alone was noninferior to gemcitabine, with a median overall survival of 9.7 and 8.8 months (P<0.001). Although the overall survival time was 10.1 months, the GS combination failed to demonstrate superiority. Additionally, the GS combination resulted in

grade 3 or higher toxicity, including neutropenia (61.2%), leukopenia (37.8%), anemia (17.2%), and thrombocytopenia (17.2%).

Owing to the potential of GS combination to prolong survival time with considerable toxicity, age of patients, non-significantly increased survival time with a fixed-dose rate of gemcitabine infusion [6,7], higher response rate upon addition of leucovorin to S-1 in gastric cancer [8], and the reimbursement policy in Taiwan; hence, we designed a modified regimen comprising gemcitabine, S-1, and leucovorin combination (GSL). According to this regimen, every 2 weeks, patients were administered intravenous gemcitabine 800 mg/m$^2$ at a rate of 10 mg/m$^2$/min on day 1, oral S-1 80–120 mg/day on days 1–7, and oral leucovorin 30 mg twice daily on days 1–7. In this report, we analyzed the feasibility of the GSL regimen and outcomes in patients with advanced or metastatic pancreatic cancer in three medical centers in Taiwan.

## Materials and methods

### Patients

This study was conducted as a retrospective review to evaluate the safety and efficacy of GSL therapy in patients with advanced or metastatic pancreatic cancer at three medical centers in Taiwan: China Medical University Hospital, National Cheng Kung University Hospital, and Kaohsiung Medical University Hospital. Patients' medical records between May 2015 and March 2019 were accessed. All data were fully anonymized before the review process. This retrospective study was approved by the Ethics Committee of the China Medical University and Hospital (CMUH107-REC1-140).

### Treatments

Patients were administered gemcitabine (800 mg/m$^2$ intravenously at a rate of 10 mg/m$^2$/min on day 1), S-1 (80–120 mg/day orally based on body surface area from days 1 to 7), and leucovorin (30 mg twice daily orally from days 1 to 7). The regimen was repeated every 2 weeks. Patients received GSL treatment until disease progression, development of unacceptable toxicity, or any reason resulting in the discontinuation of GSL therapy. Supportive management, including blood transfusion, granulocyte-stimulating factors, and analgesic drugs, were prescribed according to the Health Insurance Bureau guidelines.

### Evaluation of therapeutic efficacy and toxicity

For all patients, tumors were imaged using computed tomography or magnetic resonance imaging, before therapy and every 3 months after the initiation of GSL treatment, or as clinically required. In this study, we used the American Joint Committee on Cancer (AJCC version 8) clinical staging system for pancreatic cancer. Patients were required to have measurable lesions.

The tumor response was assessed according to the Response Evaluation Criteria in Solid Tumors (RECIST) 1.1 criteria. The toxicity profile of GSL was graded using the National Cancer Institute Common Terminology Criteria for Adverse Events, version 4.0.

### Statistical analysis

The outcome measures included overall response rate (ORR), time-to-treatment failure, progression-free survival, and overall survival. Time-to-treatment failure was defined as the time interval between the date of GSL initiation and the date of GSL termination owing to any cause, including disease progression, toxicity, withdrawal of consent, conversion to surgical intervention, or death. Progression-free survival was calculated from the date of GSL initiation

to the date of disease progression or death from any cause. Overall survival was calculated from the date of GSL initiation and the date of death or the last follow-up date. All time-to-treatment failure, progression-free survival, and overall survival were created and plotted using the Kaplan-Meier method. The Cox regression model was used for risk analysis in univariate or multivariate analyses of progression-free survival and overall survival. Factors with a $p$-value of less than 0.1 identified in the univariate analysis were included in the multivariate analysis. Statistical analysis was performed using SPSS version 18 for Windows (IBM Corporation, Armonk, New York). Data are expressed as mean ± standard deviation. All statistical tests were 2-sided, and the differences were considered statistically significant at a $p$-value less than 0.05.

## Results

### Patient characteristics

From May 2015 to March 2019, 49 patients with locally advanced or metastatic pancreatic adenocarcinoma pathologically confirmed by either biopsy or cytology were treated using the GSL regimen as the first-line treatment. The median age was 68.0 years, with a marginally higher number of females (57.1%). Table 1 presents the patient demographics and characteristics at study enrollment. Patients received a median of 8 cycles of GSL treatment, ranging between 1 and 95 cycles. Dose reduction, defined as a reduction in the dose of either gemcitabine or S-1 to less than 70% of the original dose, occurred in 14 patients (28.6%).

### Therapeutic efficacy

Among the 49 patients, 5 patients discontinued GSL treatment before the first image evaluation: 2 because of associated treatment toxicity, 2 owing to deterioration of performance, and 1 patient due to pulmonary embolism (Table 2). Of the 44 patients with follow-up image evaluations, 13 demonstrated a partial response (29.5%), 17 indicated stable disease (38.6%), and the disease control rate was 68.2%. In the intent-to-treat analysis, the ORR and disease control rates were 26.5% and 61.2%, respectively.

The median time-to-treatment failure was 5.79 months (95% C.I., 2.63–8.94), the progression-free survival was 6.94 months (95% C.I., 5.55–8.33), and overall survival time was 11.53 months (95% C.I., 9.94–13.13, Fig 1). The most common cause of treatment discontinuation was disease progression (32 patients, 65.3%).

Among the 22 patients who received second-line chemotherapy, 13 patients received liposomal irinotecan plus 5-fluorouracil, 7 received oxaliplatin-based chemotherapy, and 2 received paclitaxel (Table 1).

### Risk factor analyses for progression-free survival and overall survival

To identify the potential risk factors for patient survival, age, gender, performance status, main tumor location, disease status, cancer cell differentiation, cycles of GSL treatment, dose reduction of GSL, and second-line therapy were included in the univariate analyses of progression-free survival (Table 3) and overall survival (Table 4). Factors with $p$-value less than 0.1 identified in the univariate analysis were included in the multivariate analysis.

### Adverse effects

For all 49 patients, the toxicity profile was obtained and classified according to the CTCAE v4.0 criteria (Table 5). The most common grade 3 or higher toxicities of GSL treatment were hematological, including anemia (18.3%), neutropenia (6.1%), and thrombocytopenia (2%).

**Table 1. Baseline characteristics of patients with pancreatic adenocarcinoma treated with GSL chemotherapy.**

|  | Number | % |
|---|---|---|
| Number of patients | 49 | 100 |
| Age (year), median ± SD (minimum, maximum) | 68.0 ± 8.7 (47, 83) | |
| Gender | | |
| Male | 21 | 42.9 |
| Female | 28 | 57.1 |
| PS by ECOG score | | |
| 0 | 13 | 26.5 |
| 1 | 32 | 65.3 |
| 2 | 3 | 6.1 |
| 3 | 1 | 2.0 |
| Main tumor location | | |
| Head | 19 | 38.8 |
| Body | 16 | 32.6 |
| Tail | 14 | 28.6 |
| Tumor status at treatment | | |
| Locally advanced | 9 | 18.4 |
| Metastatic | 40 | 81.6 |
| Differentiation | | |
| Well | 3 | 6.1 |
| Moderately | 19 | 38.8 |
| Poorly | 12 | 24.5 |
| Unavailable | 15 | 30.6 |
| Cycles of GSL | | |
| Median ± SD (minimum, maximum) | 8 ± 14.4 (1, 95) | |
| <8 | 23 | 46.9 |
| ≥8 | 26 | 53.1 |
| Dose reduction[a] | | |
| No | 35 | 71.4 |
| Yes | 14 | 28.6 |
| Second- line therapy | | |
| No | 27 | 55.1 |
| Liposomal irinotecan-based | 13 | 26.5 |
| Oxaliplatin-based | 7 | 14.3 |
| Paclitaxel | 2 | 4.1 |

Abbreviations: ECOG, Eastern Cooperative Oncology Group; PS, performance status; SD, standard deviation

[a]Reduction of dose of either gemcitabine or S-1 to less than 70% of the original dose.

No patient presented with treatment-related grade 5 adverse events. Treatment discontinuation owing to toxicity was documented in three patients (6.1%).

## Discussion

The current study indicates the feasibility and safety of the GSL regimen in patients with locally advanced or metastatic pancreatic adenocarcinoma. The progression-free survival and overall survival time were consistent with results documented in previous landmark studies (Table 6).

**Table 2. Best of response for 44 evaluable patients with pancreatic adenocarcinoma treated with GSL chemotherapy and the details of 5 inevaluable patients.**

|  | Number | % |
|---|---|---|
| Complete response | 0 | 0 |
| Partial response | 13 | 29.5 |
| Stable disease | 17 | 38.6 |
| Progressive disease | 14 | 31.8 |
| Inevaluable patients |  |  |

No.1: stop chemotherapy after one course of GSL due to grade 3 anemia and grade 3 mucositis

No.2: stop chemotherapy after one course of GSL due to grade 3 anemia, grade 3 nausea and grade 3 vomiting

No.3: stop chemotherapy after one course of GSL due to poor performance

No.4: stop chemotherapy after one course of GSL due to pulmonary embolism

No.5: stop chemotherapy after one course of GSL due to poor performance

Although patients receiving a combination of S-1 (2 weeks on, 1 week off) and gemcitabine (1000 mg/m$^2$ on days 1 and 8 of a 21-day cycle) have demonstrated the longest progression-free survival among the three groups in the GEST study, they experienced a high possibility of hematological toxicities, necessitating clinical attention (Table 6). Thus, it is logical to modify the combination to reduce toxicity without compromising treatment efficacy. Koizumi et al. have suggested that a combination of S-1 and folinic acid is effective in a modified 2 weeks on / 2 weeks off schedule of S-1 administration in patients with metastatic colorectal cancer [9]. This modification demonstrated an ORR of 57% but with considerable grade 3 or higher toxicities. In a phase II trial, Li et al. have investigated a 2-week schedule of S-1 plus leucovorin (1 week on, 1 week off) in patients with metastatic colorectal cancer [10]. Notably, the 2-week regimen demonstrated better tolerability than the 4-week schedule, with grade 3 or higher toxicities such as diarrhea, stomatitis, anorexia, and neutropenia occurring in 8.3%, 8.3%, 2.8%, and 9.7% of patients, respectively. Reportedly, in a randomized phase II study in patients presenting gemcitabine-refractory advanced pancreatic cancer, a similar 2-week schedule of S-1 plus leucovorin significantly improves progression-free survival and produces a similar toxicity profile when compared with S-1 in a conventional 6-week schedule [11]. Furthermore, the enhanced antitumor effect of S-1 with leucovorin was evaluated in a randomized phase II trial for patients with advanced gastric cancer, demonstrating a response rate of 43% in the S-1 plus leucovorin group, 66% in the S-1 plus leucovorin and oxaliplatin groups, and 46% in the S-1 plus cisplatin group [8]. Based on these findings, we adopted a 2-week schedule of S-1 plus leucovorin as part of our modified GSL regimen.

Another modification of the GSL regimen is the infusion of gemcitabine (800 mg/m$^2$) at a fixed-dose rate of 10 mg/m$^2$/min. Although gemcitabine is usually administered as a 30-min infusion, there exists a rationale to infuse gemcitabine at a fixed-dose rate. Gemcitabine is converted to an active triphosphate form, 2′,2′-difluordeoxycytidine triphosphate (dFdCTP), which is incorporated into DNA. Phase I studies have shown that prolonged administration of gemcitabine at 10 mg/m$^2$/min maximizes the intracellular concentration of dFdCTP [12,13]. A dose-escalation study of fixed-dose rate gemcitabine in combination with capecitabine in advanced solid malignances has established the maximum tolerated dose of capecitabine as 500 mg/m$^2$ twice per day during days 1–14, with gemcitabine 800 mg/m$^2$ administered intravenously at 10 mg/m$^2$/min on days 1 and 8 [14]. In patients with pancreatic adenocarcinoma, a randomized phase II trial comparing different infusion schedules of gemcitabine has indicated that the fixed-dose rate infusion schedule results in a longer overall survival (8.0 vs. 5.0 months, p = 0.013), as well as improved 1-year (28.8 vs. 9.0%, p = 0.014) and 2-year survival

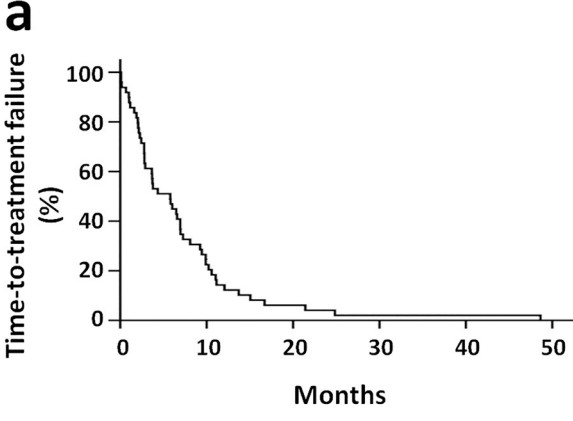

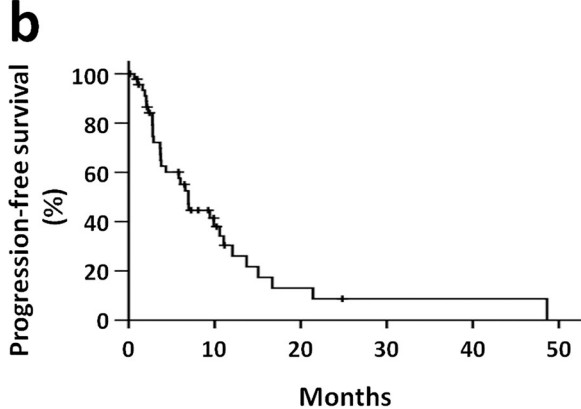

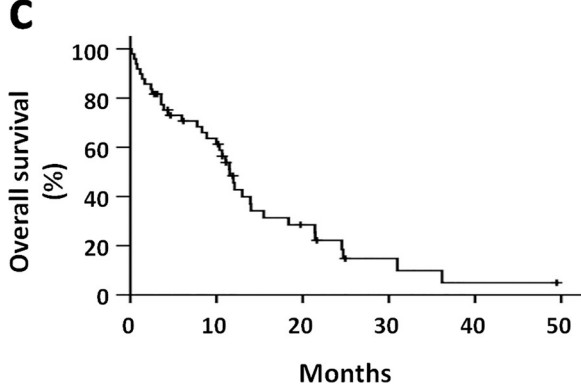

**Fig 1. Kaplan-Meier survival analysis for 49 patients.** a, time-to-treatment failure. b, progression-free survival. c, overall survival.

rates (18.3 vs. 2.2%, p = 0.007) [6]. A three-arm phase III E6201 study compared the conventional 30-min infusion of gemcitabine, fixed-dose rate infusion of gemcitabine (10 mg/m$^2$/ min), and gemcitabine plus oxaliplatin in patients with pancreatic cancer [7]. The median survival and 1-year survival rate were 4.9 months and 16% for the 30-min infusion of gemcitabine (standard), 6.2 months and 21% for fixed-dose rate infusion of gemcitabine (stratified log-rank p = 0.04), and 5.7 months and 21% for the combination (stratified log-rank p = 0.22), respectively. However, these differences failed to meet the pre-specified criteria (p<0.025) for

**Table 3. Univariate and multivariate analyses for progression-free survival (n = 49).**

| Variable | Number | Univariate Analysis | | Multivariate Analysis[a] | |
|---|---|---|---|---|---|
| | | HR (95% CI) | p Value | HR (95% CI) | p Value |
| Age (≥70 vs. <70) | 19/30 | 0.833 (0.405–1.713) | 0.620 | | |
| Gender male vs. female) | 21/28 | 0.760 (0.375–1.543) | 0.448 | | |
| PS (≥1 vs. 0) | 36/13 | 1.780 (0.796–3.980) | 0.160 | | |
| Main tumor location (body/tail vs. head) | 30/19 | 0.986 (0.481–2.022) | 0.969 | | |
| Disease status (metastatic vs. locally advanced) | 40/9 | 3.496 (1.061–11.523) | 0.040 | 5.731 (1.391–26.603) | 0.016 |
| Differentiation (poorly vs. well/moderately) | 12/22 | 1.928 (0.790–4.706) | 0.150 | | |
| GSL cycles (≥8 vs. <8) | 26/23 | 0.075 (0.029–0.196) | <0.001 | 0.037 (0.009–0.146) | <0.001 |
| Dose reduction of GSL[b] (yes vs. no) | 14/35 | 2.021 (0.994–4.108) | 0.052 | 1.879 (0.905–3.900) | 0.091 |
| Second-line therapy (yes vs. no) | 22/27 | 1.415 (0.688–2.911) | 0.346 | | |

Abbreviations: CI, confidence interval; GSL, regimen in the current study; HR, hazard ratio; PS, performance status.

[a]Factors with a p value less than 0.1 identified in univariate analysis were included in multivariate analysis.

[b]Reduction of dose of either gemcitabine or S-1 to less than 70% of the original dose.

statistical significance. Additionally, several patients experienced grade 3/4 neutropenia (59%) and thrombocytopenia (33%) in the fixed-dose rate gemcitabine arm. Consequently, to reduce hematologic toxicities, we investigated a 20% lower gemcitabine dose (800 mg/m$^2$) at a fixed-dose rate. The administration of biweekly gemcitabine of 800 mg/m$^2$ at a fixed-dose rate has been adapted in our previous trials for biliary tract or pancreatic cancers. Gemcitabine of 800 mg/m$^2$ at 10 mg/m$^2$/min plus oxaliplatin 85 mg/m$^2$ were active for patients with advanced biliary tract cancer [15]. Furthermore, a biweekly SLOG regimen (gemcitabine of 800 mg/m$^2$ at 10 mg/m$^2$/min followed by 85 mg/m$^2$ oxaliplatin on day 1 plus S-1 and leucovorin twice daily on days 1–7) has revealed promising activity and safety profiles for metastatic pancreatic adenocarcinoma in a phase II trial [16].

A major advantage of our GSL regimen is its relatively lower toxicities when compared with previous standard regimens (Table 4). The most common grade 3 or higher toxicities experienced with GSL were anemia in 18.3%, and neutropenia in 6.1% of patients. A chemotherapeutic regimen with equal effectiveness and attenuated toxicities is especially crucial for improving outcomes in elderly patients. Compared with the previous studies, our patients were older, with a median age of 68 years, consistent with the data (male with median age 65 years, female with median age 69 years) obtained from the Health Promotion Administration, Ministry of Health and Welfare, Taiwan [17]. In our cohort, 39%, 20%, and 41% of patients were aged <65 years, 65–70 years, and ≥ 70 years, respectively.

Moreover, apart from attempting to reduce treatment-related toxicities, another consideration to design the GSL regimen was that the National Health Insurance system in Taiwan did not reimburse nab-paclitaxel, oxaliplatin, and irinotecan prior to 2020. Instead, in Taiwan, gemcitabine or S-1 are extensively used in patients with pancreatic, gastric, or biliary tract cancer.

Recently, Saito and colleagues have reported a phase II study using a GSL regimen similar to our regimen in Japanese patients with advanced pancreatic cancer [18]. Patients were treated with gemcitabine 1000 mg/m$^2$ over 30 min on day 1, and S-1 40 mg/m$^2$ and leucovorin 25 mg orally administered twice a day on days 1–7; each cycle was repeated every 2 weeks. Our GSL regimen differs from Saito's in the fixed-dose rate and lower gemcitabine dose based on

**Table 4. Univariate and multivariate analyses for overall survival (n = 49).**

| Variable | Number | Univariate Analysis | | Multivariate Analysis[a] | |
|---|---|---|---|---|---|
| | | HR (95% CI) | *p* Value | HR (95% CI) | *p* Value |
| Age (≥70 vs. <70) | 19/30 | 0.928 (0.447–1.925) | 0.841 | | |
| Gender male vs. female) | 21/28 | 0.889 (0.457–1.731) | 0.730 | | |
| PS (≥1 vs. 0) | 36/13 | 4.747 (1.884–11.961) | 0.001 | 4.337 (1.619–11.622) | 0.004 |
| Main tumor location (body/tail vs. head) | 30/19 | 0.809 (0.416–1.574) | 0.533 | | |
| Disease status (metastatic vs. locally advanced) | 40/9 | 0.907 (0.390–2.110) | 0.821 | | |
| Differentiation (poorly vs. well/moderately) | 12/22 | 1.318 (0.524–3.314) | 0.557 | | |
| GSL cycles (≥8 vs. <8) | 26/23 | 0.227 (0.106–0.485) | <0.001 | 0.183 (0.080–0.417) | <0.001 |
| Dose reduction of GSL (yes vs. no) | 14/35 | 1.781 (0.894–3.548) | 0.101 | | |
| Second-line therapy (yes vs. no) | 22/27 | 0.477 (0.239–0.951) | 0.036 | 0.621 (0.283–1.363) | 0.235 |

Abbreviations: CI, confidence interval; GSL, regimen in the current study; HR, hazard ratio; PS, performance status.

[a]Factors with a *p* value less than 0.1 identified in univariate analysis were included in multivariate analysis.

[b]Reduction of dose of either gemcitabine or S-1 to less than 70% of the original dose.

the rationale stated above. In the study by Saito, 19 patients with locally advanced cancer and 30 patients with metastatic pancreatic cancer were enrolled. The ORR, disease control rate, median progression-free survival, and overall survival were 32.7%, 87.8%, 10.8 months, and 20.7 months, respectively. The reported grade 3 or higher toxicities included neutropenia (22.4%) and stomatitis (14.3%). Notably, 28 patients in this study cohort received second-line chemotherapy, with 12 patients receiving nab-paclitaxel plus gemcitabine, 6 patients receiving FOLFIRINOX or modified FOLFIRINOX, 6 receiving gemcitabine plus S-1, 2 receiving irinotecan, and 2 patients receiving other drugs. The remarkable overall survival time observed could be attributed to the salvage chemotherapeutic regimens previously not reimbursed by the Health Insurance Bureau in Taiwan.

Our study has several limitations. First, this was a retrospective study with unavoidable bias. Second, the number of patients in our study was small. However, our results suggest that this GSL regimen is feasible with a relative safety profile. To further validate this observation, a prospective trial evaluating the same regimen in elderly patients with locally advanced or metastatic pancreatic adenocarcinoma is currently ongoing in Taiwan.

In conclusion, our modified GSL combination therapy is feasible as the first-line treatment for advanced and metastatic pancreatic cancer, with tolerable toxicities. The modified GSL

**Table 5. Adverse effects of 49 patients with pancreatic adenocarcinoma treated with GSL chemotherapy.**

| | Grade [n (%)] | | | | |
|---|---|---|---|---|---|
| | 0 | 1 | 2 | 3 | 4 |
| Neutropenia | 37 (75.5) | 4 (8.2) | 5 (10.2) | 1 (2.0) | 2 (4.1) |
| Anemia | 6 (12.2) | 10 (20.4) | 24 (49.0) | 8 (16.3) | 1 (2.0) |
| Thrombocytopenia | 32 (65.3) | 6 (12.2) | 10 (20.4) | 1 (2.0) | 0 |
| Renal dysfunction | 26 (53.1) | 11 (22.4) | 12 (24.5) | 0 | 0 |
| Nausea | 41 (83.7) | 0 | 6 (12.2) | 2 (4.1) | 0 |
| Vomiting | 45 (91.8) | 0 | 3 (6.1) | 1 (2.0) | 0 |
| Mucositis | 43 (87.7) | 1 (2.0) | 3 (6.1) | 2 (4.1) | 0 |
| Diarrhea | 47 (95.9) | 2 (4.1) | 0 | 0 | 0 |

**Table 6. Outcomes and grade 3 or higher toxicities of regimens used for patients with locally advanced or metastatic pancreatic adenocarcinoma.**

| Regimen | FOLFIRINOX (Conroy) [2] | nab-paclitaxel+gemcitabine (von Hoff) [3] | S-1 (Ueno) [5] | S-1+gemcitabine (Ueno) [5] | GSL (present, intent-to-treat) |
|---|---|---|---|---|---|
| Patients (n) | 171 | 431 | 280 | 275 | 49 |
| Median age (yr) | 61 | 62 | ≥ 65 (48.2%) | ≥ 65 (50.2%) | 68 |
| ORR (%) | 31.6 (24.7–39.1) | 23 (19–27) | 21.0 (16.1–26.6) | 29.3 (23.7–35.5) | 26.5 |
| TTF (month) | N.A. | 5.1 (4.1–5.5) | N.A. | N.A. | 5.8 (2.6–8.9) |
| PFS (month) | 6.4 (5.5–7.2) | 5.5 (4.5–5.9) | 3.8 (2.9–4.2) | 5.7 (5.4–6.7) | 6.9 (5.5–8.3) |
| OS (month) | 11.1 (9.0–13.1) | 8.5 (7.9–9.5) | 9.7 (7.6–10.8) | 10.1 (9.0–11.2) | 11.5 (9.9–13.1) |
| 1y-OS rate (%) | 48.4 | 35 (30–39) | 38.7 | 40.7 | |
| Neutropenia (%) | 45.7 | 38 | 8.8 | 62.2 | 6.1 |
| Anemia (%) | 7.8 | 13 | 9.6 | 17.2 | 18.3 |
| Thrombocytopenia (%) | 9.1 | 13 | 1.5 | 17.2 | 2.0 |
| Vomiting (%) | 14.5 | N.A. | 1.5 | 4.5 | 2.0 |
| Diarrhea (%) | 12.7 | 6 | 5.5 | 4.5 | 0 |
| Mucositis (%) | N.A. | N.A. | 0.7 | 2.2 | 4.1 |

Abbreviations: N.A., not available; ORR, overall response rate; OS, overall survival; PFS, progression-free survival; TTF, time-to-treatment failure.

therapy can be a promising and affordable choice of treatment in patients with advanced and metastatic pancreatic cancer in Taiwan; however, further prospective trials remain crucial.

## Supporting information

**S1 File. PC patient using GSL all minimal data set-20201130-1.**
(XLSX)

## Author Contributions

**Conceptualization:** Chang-Fang Chiu, Li-Tzong Chen, Li-Yuan Bai.

**Data curation:** Chia-Yu Chen, Shih-Hsin Liang, Yung-Yeh Su, Nai-Jung Chiang, Hui-Ching Wang.

**Formal analysis:** Chia-Yu Chen, Shih-Hsin Liang, Yung-Yeh Su, Nai-Jung Chiang, Li-Tzong Chen.

**Funding acquisition:** Chang-Fang Chiu, Li-Yuan Bai.

**Investigation:** Chia-Yu Chen, Shih-Hsin Liang, Yung-Yeh Su, Hui-Ching Wang, Li-Yuan Bai.

**Resources:** Chang-Fang Chiu, Li-Tzong Chen.

**Writing – original draft:** Shih-Hsin Liang, Yung-Yeh Su, Nai-Jung Chiang, Hui-Ching Wang, Li-Yuan Bai.

**Writing – review & editing:** Chang-Fang Chiu, Li-Tzong Chen.

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
