## [Decision Letter · Decision Letter 0]

17 Nov 2020

PONE-D-20-24681

Modified Gemcitabine, S-1, and Leucovorin Combination for Patients with Newly Diagnosed Advanced or Metastatic Pancreatic Adenocarcinoma: A Multi-center Retrospective Study in Taiwan

PLOS ONE

Dear Dr. Li-Yuan Bai,

Thank you for submitting your manuscript to PLOS ONE. After careful consideration, we feel that it has merit but does not fully meet PLOS ONE’s publication criteria as it currently stands. Therefore, we invite you to submit a revised version of the manuscript that addresses the points raised during the review process.

We reccomend to address specific comments of the Reviewers with specific attention to mitigate the impact of the retrospective study desgin by adding additional clinical co-variates to the final analysis. According to the Reviewers suggestion English language should be revised.

We look forward to receiving your revised manuscript.

Kind regards,

Erika Cecchin

Academic Editor

PLOS ONE

Journal Requirements:

'The study was approved by the Local Ethics Committee (CMUH107-REC1-140) with a waiver of informed consent.'

(a) Please amend your current ethics statement to include the full name of the ethics committee/institutional review board(s) that approved your specific study.  

(b) Once you have amended this/these statement(s) in the Methods section of the manuscript, please add the same text to the “Ethics Statement” field of the submission form (via “Edit Submission”).

3. In the ethics statement in the manuscript and in the online submission form, please provide additional information about the patient records used in your retrospective study, including: a) whether all data were fully anonymized before you accessed them and b) the date range (month and year) during which patients' medical records were accessed.

4. To comply with PLOS ONE submission guidelines, in your Methods section, please provide additional information regarding your statistical analyses. For more information on PLOS ONE's expectations for statistical reporting, please see https://journals.plos.org/plosone/s/submission-guidelines.#loc-statistical-reporting.

Reviewers' comments:

Reviewer's Responses to Questions

**Comments to the Author**

1. Is the manuscript technically sound, and do the data support the conclusions?

Reviewer #1: Yes

Reviewer #2: Yes

Reviewer #3: Yes

2. Has the statistical analysis been performed appropriately and rigorously? 

Reviewer #1: Yes

Reviewer #2: Yes

Reviewer #3: No

3. Have the authors made all data underlying the findings in their manuscript fully available?

Reviewer #1: Yes

Reviewer #2: Yes

Reviewer #3: Yes

4. Is the manuscript presented in an intelligible fashion and written in standard English?

Reviewer #1: Yes

Reviewer #2: Yes

Reviewer #3: Yes

5. Review Comments to the Author

Reviewer #1: This study aimed to retrospectively evaluate the feasibility of a modified chemotherapeutic regimen using the combination of gemcitabine plus the oral fluoropyrimidine S-1, as first-line treatment of locally advanced or metastatic pancreatic adenocarcinoma in Taiwan before the reimbursement of nab-paclitaxel. Among the 44 evaluable patients. partial response rate and disease control rate were 29.5% and 68.2%, respectively; progression-free survival and overall survival were 6.94 and 11.53 months, respectively. The most common grade 3 or higher toxicities were anemia (18.3%) and neutropenia (6.1%). These results suggest that the investigating therapeutic regimen could be feasible and relatively safe. However, some limitations in this study should be considered, including its retrospective nature and the limited sample size.

The present manuscript is well-written and clear regarding its endpoints; however, it should be subjected to some minor changes, as it follows:

-In the title please modify Advanced in Locally Advanced.

-In the paragraph "Introduction": on line 77 please correct trial to trials; on line 82 please specify that the considerable

toxicities are related to FOLFIRINOX; on line 89 however should be substituted with despite of modified schedule; on line 113

please specify that the reference [8] is referred to gastric and not pancreatic cancer.

-In the paragraph “Discussion” authors should better explain the reasons why gemcitabine was given at a 20% lower dose

(800 mg instead of 1000 mg/m2) and with a modified schedule (on day 1 every 2 weeks instead of days 1 and 8 every 3

weeks). Indeed similar modifications of dosage and schedule could have affected safety and toxicity results; authors should

mention and argue the point.

Reviewer #2: This is a retrospective study of a chemotherapy regimen GSL for patients with locally advanced and/or metastatic pancreas cancer. There are only 49 patients in the study. The toxicity was manageable so the chemotherapy was reasonably well tolerated. However, the efficacy was minimal as the median survival was 10 months and the median progression-free survival was 8 months. The regimen has minimal efficacy.

Reviewer #3: The manuscript entitled ‘Modified Gemcitabine, S-1, and Leucovorin Combination for Patients with Newly Diagnosed Advanced or Metastatic Pancreatic Adenocarcinoma: A Multi-center Retrospective Study in Taiwan’ was aimed at evaluating efficacy and toxicity of first-line gemcitabine administered at a fixed-dose rate in combination with the oral fluoropyrimidine S-1 and folinic acid in late stage pancreatic cancer patients.

As the authors well highlight in the manuscript, more active treatments are currently available also in Taiwan, however this study has some value and results showed a substantial activity of the chemotherapeutic combination.

Unfortunately, the retrospective nature of this study represents an unfavorable aspect. Thus, in order to maximally exploit this study and to provide the reader with more relevant information, the following aspects (see below) have to be satisfied.

Main comments

Results are limited to the description of survival parameters, objective response, toxicity.

In order to better characterize the response (efficacy/toxicity) of this setting of patients in respect to the study combination, a proper statistical analysis (univariate and multivariate analysis) of relationships between efficacy (survival parameters - PFS and OS; objective response) or toxicity and clinical/pathological characteristics included in Table 1 should be performed.

The statistical analysis should also include as variables to be considered the administered cycles and dose reductions (i.e. by subdividing patients according to at least two groups) and second line treatments. Please, add in table 1, accordingly, the mean number of administered cycles (and range), dose reductions and second line treatments.

English language should be revised by a mother tongue person.

Minor comments

Please, include in Table 3 also the absolute numbers of patients and not only percentages.

6. PLOS authors have the option to publish the peer review history of their article (what does this mean?). If published, this will include your full peer review and any attached files.

Reviewer #1: No

Reviewer #2: No

Reviewer #3: No

---

## [Author Response · Author response to Decision Letter 0]

6 Dec 2020

Editor comments:

The whole manuscript has been revised to meet the PLOS ONE’s style requirements.

2. Thank you for including your ethics statement: 'The study was approved by the Local Ethics Committee (CMUH107-REC1-140) with a waiver of informed consent.' (a) Please amend your current ethics statement to include the full name of the ethics committee/institutional review board(s) that approved your specific study. (b) Once you have amended this/these statement(s) in the Methods section of the manuscript, please add the same text to the “Ethics Statement” field of the submission form (via “Edit Submission”).

The full name of the ethics committee has been added in the paragraph of “Materials and Methods” (line 133) and the “Ethics Statement” field online.

3. In the ethics statement in the manuscript and in the online submission form, please provide additional information about the patient records used in your retrospective study, including: a) whether all data were fully anonymized before you accessed them and b) the date range (month and year) during which patients' medical records were accessed.

Per suggestion, the relevant statement and date range for data collection are added in “Materials and Methods” (line 136-137).

4. To comply with PLOS ONE submission guidelines, in your Methods section, please provide additional information regarding your statistical analyses. For more information on PLOS ONE's expectations for statistical reporting, please see https://journals.plos.org/plosone/s/submission-guidelines.#loc-statistical-reporting.

We have revised the relevant statement for statistical analysis (line 170-174).

We have confirmed that the ethics statement is listed in only the Materials and Methods section.

Reviewers' comments:

Reviewer #1:

Q1: In the title please modify Advanced in Locally Advanced.

It has been revised accordingly (line 3).

Q2: In the paragraph "Introduction": on line 77 please correct trial to trials; on line 82 please specify that the considerable toxicities are related to FOLFIRINOX; on line 89 however should be substituted with despite of modified schedule; on line 113 please specify that the reference [8] is referred to gastric and not pancreatic cancer.

Thank you for suggestions. All points are revised accordingly (line 80, 85, 93, and 117).

Q3:In the paragraph “Discussion” authors should better explain the reasons why gemcitabine was given at a 20% lower dose (800 mg instead of 1000 mg/m2) and with a modified schedule (on day 1 every 2 weeks instead of days 1 and 8 every 3 weeks). Indeed similar modifications of dosage and schedule could have affected safety and toxicity results; authors should mention and argue the point. 

We thank you the great question. The rationale and background of using biweekly gemcitabine at 800 mg/m2 are added in Discussion (line 254-261, 276-283).

Reviewer #2:

Q1: This is a retrospective study of a chemotherapy regimen GSL for patients with locally advanced and/or metastatic pancreas cancer. There are only 49 patients in the study. The toxicity was manageable so the chemotherapy was reasonably well tolerated. However, the efficacy was minimal as the median survival was 10 months and the median progression-free survival was 8 months. The regimen has minimal efficacy.

Thank you for the question. The major point of this retrospective study is to show the relatively lower toxicities of GSL regimen compared with previous standard regimens without compromising the activity for patients with pancreatic cancer. The median progression-free survival and overall survival time was 6.94 and 11.53 months, and 5.7 and 10.1 months in our study and in the phase III GEST study (GS arm), respectively. Because of the modest activity and safety profiles, a prospective trial evaluating the same regimen in elderly patients with locally advanced or metastatic pancreatic adenocarcinoma is currently ongoing in Taiwan.

Reviewer #3:

Q1: Results are limited to the description of survival parameters, objective response, toxicity. In order to better characterize the response (efficacy/toxicity) of this setting of patients in respect to the study combination, a proper statistical analysis (univariate and multivariate analysis) of relationships between efficacy (survival parameters - PFS and OS; objective response) or toxicity and clinical/pathological characteristics included in Table 1 should be performed.

Thank you for great suggestion. The univariate and multivariate analyses of risk factors for progression-free survival and overall survival have been performed and been shown in new Table 3 and 4. 

Q2: The statistical analysis should also include as variables to be considered the administered cycles and dose reductions (i.e. by subdividing patients according to at least two groups) and second line treatments. Please, add in table 1, accordingly, the mean number of administered cycles (and range), dose reductions and second line treatments.

The information of administration cycles, dose reduction and second line therapy are added in Table 1 and content (line 201-203). We also include the administration cycle, dose reduction and second line therapy as parameters in the univariate and multivariate analyses for progression-free survival and overall survival (Table 3 and 4).

Q3: English language should be revised by a mother tongue person.

Thank you for your suggestion. We have revised and polished the manuscript with the help of an English editing company Editage for its linguistic assistance. The proof of the service provided by Editage is attached here for your reference.

Q4: Please, include in Table 3 also the absolute numbers of patients and not only percentages. 

The numbers and percentages of patients have been added in new Table 5.

---

## [Decision Letter · Decision Letter 1]

11 Dec 2020

Modified Gemcitabine, S-1, and Leucovorin Combination for Patients with Newly Diagnosed Locally Advanced or Metastatic Pancreatic Adenocarcinoma: A Multi-center Retrospective Study in Taiwan

PONE-D-20-24681R1

Dear Dr. Li-Yuan Bai,

We’re pleased to inform you that your manuscript has been judged scientifically suitable for publication and will be formally accepted for publication once it meets all outstanding technical requirements.

Kind regards,

Erika Cecchin

Academic Editor

PLOS ONE

Additional Editor Comments (optional):

The Authors addressed the criticisms raised by the Reviewers and better addressed their study limitations in the paper discussion.

Reviewers' comments:

Reviewer's Responses to Questions

**Comments to the Author**

1. If the authors have adequately addressed your comments raised in a previous round of review and you feel that this manuscript is now acceptable for publication, you may indicate that here to bypass the “Comments to the Author” section, enter your conflict of interest statement in the “Confidential to Editor” section, and submit your "Accept" recommendation.

Reviewer #1: All comments have been addressed

Reviewer #2: All comments have been addressed

2. Is the manuscript technically sound, and do the data support the conclusions?

Reviewer #1: (No Response)

Reviewer #2: Yes

3. Has the statistical analysis been performed appropriately and rigorously? 

Reviewer #1: (No Response)

Reviewer #2: Yes

4. Have the authors made all data underlying the findings in their manuscript fully available?

Reviewer #1: (No Response)

Reviewer #2: Yes

5. Is the manuscript presented in an intelligible fashion and written in standard English?

Reviewer #1: (No Response)

Reviewer #2: Yes

6. Review Comments to the Author

Reviewer #1: (No Response)

Reviewer #2: I think that the authors have carefully considered the criticisms of each reviewer and revised the manuscript accordingly. It is now acceptable for publication

7. PLOS authors have the option to publish the peer review history of their article (what does this mean?). If published, this will include your full peer review and any attached files.

Reviewer #1: **Yes: **Michela Guardascione

Reviewer #2: No

---

## [Editor Report · Acceptance letter]

17 Dec 2020

PONE-D-20-24681R1 

Modified Gemcitabine, S-1, and Leucovorin Combination for Patients with Newly Diagnosed Locally Advanced or Metastatic Pancreatic Adenocarcinoma: A Multi-center Retrospective Study in Taiwan 

Dear Dr. Bai:

I'm pleased to inform you that your manuscript has been deemed suitable for publication in PLOS ONE. Congratulations! Your manuscript is now with our production department. 

Kind regards, 

on behalf of

Dr. Erika Cecchin 

Academic Editor

PLOS ONE